# Independent and Interactive Effects of Precipitation Intensity and Duration on Soil Microbial Communities in Forest and Grassland Ecosystems of China: A Meta-Analysis

**DOI:** 10.3390/microorganisms13081915

**Published:** 2025-08-17

**Authors:** Bo Hu, Wei Li

**Affiliations:** 1College of Soil and Water Conservation, Southwest Forestry University, Kunming 650224, China; hubo684@gmail.com; 2Key Laboratory of Ecological Environment Evolution and Pollution Control in Mountainous & Rural Areas of Yunnan Province, Kunming 650224, China; 3Zhanyi Karst Ecosystem Observation and Research Station, Qujing 650224, China

**Keywords:** precipitation gradient, temporal scale, soil microbial biomass, microbial diversity, enzyme activity, forest ecosystem, grassland ecosystem, meta-analysis

## Abstract

Altered precipitation regimes, both in intensity and duration, can profoundly influence the structure and function of soil microbial communities, yet the patterns and drivers of these responses remain unclear across ecosystem types. Here, using data exclusively from 101 field experiments conducted in China (yielding 695 observations), we investigated the impacts of altered precipitation on soil microbial biomass, diversity, and enzymatic activity in forest and grassland ecosystems. Soil microbial biomass carbon (MBC) and nitrogen (MBN) increased in response to precipitation addition, particularly in grasslands, but they decreased under reduced precipitation, with the decline being more pronounced in forests. The magnitude and duration of precipitation manipulation significantly influenced these effects, with moderate and long-term changes producing divergent responses. Bacterial diversity was largely unaffected by all precipitation treatments, whereas fungal diversity decreased significantly under intense and short-term reductions in precipitation. Enzyme activities exhibited the following element-specific patterns: carbon- and phosphorus-cycling enzymes and antioxidant enzymes were suppressed by precipitation reduction, especially in grasslands, while nitrogen-cycling enzymes showed no consistent response. Moreover, microbial responses were significantly shaped by environmental factors, including mean annual temperature (MAT), mean annual precipitation (MAP), and elevation. Our region-specific analysis highlights precipitation-driven microbial dynamics across China’s diverse climatic and ecological conditions. These findings demonstrate that soil microbial communities respond asymmetrically to precipitation changes, with responses shaped by both ecosystem type and climatic context, underscoring the need to account for environmental heterogeneity when predicting belowground feedback to climate change.

## 1. Introduction

Global climate change is accelerating, and recent reports by the Intergovernmental Panel on Climate Change (IPCC) indicate that the rates of warming and sea-level rise in the 21st century are unprecedented in the context of the past 2000 years, contributing to more frequent and intense extreme precipitation and drought events [1]. Soil microbial communities, including bacteria and fungi, are critical drivers of terrestrial biogeochemical cycles, particularly through their regulation of carbon (C) and nitrogen (N) turnover in response to altered water availability, which affects their biomass and metabolic activity [2,3]. Microbial growth and activity are highly sensitive to changes in water availability, as evidenced by the increased MBC and MBN under elevated precipitation in arid ecosystems [4]. Changes in precipitation patterns primarily influence microbial biomass, diversity, and community structure by altering soil water availability [5,6]. Dry conditions suppress microbial activity by reducing soil water availability, while increased precipitation enhances microbial biomass and nutrient availability in the topsoil [7,8]. Precipitation addition generally stimulates microbial growth and activity by enhancing soil water availability and nutrient diffusion, whereas precipitation reduction tends to suppress microbial respiration and alter community structure [9,10].

Moderate increases in precipitation are associated with enhanced microbial community complexity and diversity, whereas reduced precipitation can simplify microbial networks and lower diversity, likely due to water scarcity and limited nutrient availability [11]. Precipitation exclusion diminishes microbial function by reducing both labile and stable carbon pools, although forest types vary in their resilience to water stress [12]. Microbial responses to precipitation can also be indirectly influenced by the associated plant dynamics such as litter inputs and rhizosphere changes [13]. Soil properties, such as texture and organic matter content, can mediate microbial responses to precipitation by influencing water retention and nutrient distribution [14]. However, microbial responses to altered precipitation may differ substantially between ecosystems such as grasslands and forests due to contrasting biogeochemical conditions [15]. A growing number of field studies have shown that both increases and decreases in precipitation can significantly reshape soil microbial biomass, enzymatic activity, and community composition across forest and grassland ecosystems [16,17,18]. Existing studies often overlook biome-level heterogeneity, where microbial communities in grasslands and forests respond differently to identical treatments due to differences in soil properties, climatic history, or microbial baseline composition [19].

Although precipitation is recognized as a key driver of microbial dynamics in arid systems, few studies distinguish between microbial responses across varying precipitation intensities and durations [20]. Many experiments treat irrigation or drought as binary treatments, often overlooking whether moderate versus extreme events trigger threshold responses in microbial community structure or enzyme activity [21,22,23]. Some experimental designs lack the systematic investigation of precipitation intensity and duration gradients, instead relying on binary treatments that fail to identify threshold or nonlinear responses [24,25]. Despite growing evidence that temporal scale—from short- to long-term precipitation changes—affects microbial community responses, comparative studies spanning multiple durations remain limited [26,27,28]. Meanwhile, similar irrigation or drought treatments often lead to contrasting microbial responses, including increases, decreases, or no changes in biomass [29,30]. Although evidence has increasingly shown that altered precipitation regimes affect microbial communities in semi-arid grasslands, substantial uncertainties persist regarding the consistency and underlying drivers of these responses across different ecosystems [31,32,33]. This research gap calls for a comprehensive cross-biome meta-analysis to unify microbial response patterns under altered precipitation regimes [34].

Although some meta-analyses have assessed the effects of precipitation changes on soil microbial communities at the global scale, syntheses specifically focused on China remain scarce [35,36]. Many field experiments conducted in China, particularly those published in Chinese-language journals, have yet to be included in global datasets [37]. Due to the large spatial heterogeneity inherent in global-scale analyses, it is often challenging to accurately isolate and interpret microbial responses to precipitation. As a region particularly vulnerable to extreme droughts and rainfall events driven by climate change, China has implemented extensive precipitation manipulation experiments, providing a valuable basis for meta-analysis. A regionally focused synthesis therefore provides a necessary, context-specific perspective on soil microbial responses to altered precipitation and follows precedents established in the related ecological research [38,39].

Microbial responses to precipitation changes are highly context-dependent, shaped by region-specific environmental conditions and biological interactions [40]. Baseline climate conditions such as MAT can amplify or buffer the effects of precipitation changes on microbial communities [41,42,43]. These limitations underscore the need for a systematic synthesis to elucidate how soil microbial biomass, diversity, and activity respond to varying precipitation intensities and durations across different ecosystems. The present study aims to address the following two unresolved questions: (1) How do changes in precipitation intensity and duration independently and interactively influence soil microbial biomass, activity, and diversity across China’s forest and grassland ecosystems? (2) What roles do ecosystem-specific drivers play in shaping these responses under complex global change scenarios?

## 2. Materials and Methods

### 2.1. Data Collection

All data in this study were retrieved from the Web of Science (https://www.webofscience.com/, accessed on 16 November 2024) and China National Knowledge Infrastructure (CNKI) (https://www.cnki.net/, accessed on 17 November 2024).

The following keywords were used for the literature search: China; Forest; Grassland; Precipitation increase/addition; Precipitation decrease/drought; Microbial biomass carbon (MBC); Microbial biomass nitrogen (MBN); Soil or microbial enzyme activity; Microbial diversity or Shannon index; Fungi or fungal community; and Bacteria or bacterial community. We have collected peer-reviewed papers published up until November 2024. The articles were screened according to the following criteria: (1) The figures or tables in the article must contain extractable data, including the mean, sample size, standard deviation, or standard error. (2) Only studies investigating the effects of changes in precipitation (increase or decrease) on soil microorganisms were included; studies involving multiple global change factors were excluded. (3) Only field experiments conducted in forest and grassland ecosystems within China were considered, excluding studies from other countries, other ecosystem types, or laboratory experiments. (4) If a single article included multiple study sites, each site was treated as an independent observation. When the same dataset appeared in multiple publications originating from the same study, the data were recorded only once. Based on the precipitation manipulation intensity and duration reported in the collected literature, precipitation changes were categorized as follows: Precipitation intensity manipulations were categorized as slight (SPI/SPD, change within ±≤30%), moderate (MPI/MPD, ±30% to ±60%), and extreme (EPI/EPD, ±≥60%). Precipitation duration was classified into short-term (STPI/STPD, ≤2 years), medium-term (MTPI/MTPD, 2–4 years), and long-term (LTPI/LTPD, ≥4 years). Since there is currently no universally accepted standard for categorizing precipitation manipulation intensity and duration, we defined the thresholds based on the distribution of values in our dataset and their ecological relevance. The use of ≤2 years as the short-term cutoff helped balance sample sizes across categories. Although no unified scheme exists, a similar classification logic has been adopted in previous ecological meta-analyses, supporting the general validity of our approach [44].

Data reported in the text, tables, and Appendix A were extracted directly, while data available only in figures were digitized using GetData Graph Digitizer (version 2.26) to ensure accurate retrieval.

Based on the above criteria, a total of 101 publications were collected, yielding 695 observations. Among these, 57 studies were published within the past five years. These observations were categorized as follows: MBC, determined using the chloroform fumigation-extraction method (269 observations from 65 publications), and MBN (148 observations from 46 publications); bacterial Shannon diversity index (61 observations from 16 publications) and fungal Shannon diversity index (55 observations from 14 publications); and soil carbon-cycling enzyme activities (75 observations from 12 publications), nitrogen-cycling enzyme activities (9 observations from 4 publications), phosphorus-cycling enzyme activities (34 observations from 12 publications), and antioxidant enzyme activities (44 observations from 9 publications). The geographic distribution of the sampling sites is shown in Figure 1.

In addition, to explore the influence of geographic environmental conditions on soil microbial communities, we collected information on the latitude, longitude, MAT, MAP, and altitude of each experimental site. If MAT, MAP, and altitude were not reported in the original publications, these variables were extracted from the WorldClim2 database based on the geographic coordinates of the site [45].

### 2.2. Calculation of the Treatment Effect Size

This meta-analysis quantified the effects of precipitation changes on soil microbial community structure and function by calculating the natural log-transformed response ratio (*lnRR*) as a standardized effect size. The effect size of a single precipitation treatment on soil microbial communities was calculated as follows:(1)$$\ln RR=\ln\frac{X_{t}}{X_{c}}=\ln X_{t}-\ln X_{c}$$
where lnR denotes the effect size of a single precipitation treatment, *X_t_* denotes the average value in the treatment, and *X_c_* denotes the average value in the control [46]. X_t_ and X_c_ represent the mean values of microbial indicators (e.g., MBC, MBN, diversity indices, and enzyme activities) under altered precipitation treatment and ambient control conditions, respectively, as reported in the original studies.

For data analysis purposes, some reported standard errors (SEs) were converted to standard deviations (SDs). The conversions were performed as follows:(2)SD=SE×√n
where n denotes sample size.

The within-group variance(*v*) for each treatment group was calculated as follows:(3)$$v=\frac{S_{t}^{2}}{N_{t}X_{t}^{2}}+\frac{S_{c}^{2}}{N_{c}X_{c}^{2}}$$
where *N_t_* and *N_c_* denote the sample sizes of the treatment and control groups, respectively; *St* and *Sc* refer to the standard deviations of the treatment and control groups; and *X_t_* and *X_c_* indicate the corresponding mean values [47].

The weight assigned to each individual study was calculated using the following formula.(4)$$w_{i}^{*}=\frac{1}{v_{i}+r^{2}}$$
where $w_{i}^{*}$ denotes the assigned weight, while $v_{i}$ and $r^{2}$ correspond to the within-study variance and between-study variance, respectively.

A random-effects model was applied to estimate the mean effect size in the meta-analysis, as calculated by the following equation:(5)$$\bar{Y}=\frac{\sum_{i=1}^{k}w_{i}^{*}y_{i}}{\sum_{i=1}^{k}w_{i}^{*}}$$
where $\bar{Y}$ represents the average effect size; y_i_ denotes the effect size of an individual study.

For ease of interpretation, the lnRR and its corresponding confidence intervals were converted into percentage change as follows:(6)$$CI=(e^{\ln RR}-1)\times 100\%$$

### 2.3. Data Processing and Analysis

A random-effects model was used in this study to conduct the meta-analysis, calculating both individual effect sizes and overall mean effect sizes. When the 95% confidence interval of the mean effect size does not overlap with zero, the precipitation treatment is considered to have a statistically significant effect on the biomass, diversity, and activity of soil microbial communities. A mean effect size greater than zero indicates a positive effect of precipitation treatment on microbial biomass, diversity, and enzyme activities, while a value less than zero indicates a negative effect. Effect size calculations were performed using MetaWin 2.1, and correlation analyses between environmental factors and microbial biomass, diversity, and enzyme activities were conducted and visualized using GraphPad Prism 10.

## 3. Results

### 3.1. Effects of Precipitation Changes on MBC and MBN

Precipitation addition significantly increased MBC by 11.96% overall (95% confidence interval, 7.13% to 16.79%), while reduction had a suppressive effect, with the two responses differing significantly (*p* < 0.001). In forests, MBC decreased under precipitation reduction but showed no response to addition. In grasslands, an increase in precipitation significantly raised MBC by 15.87% (95CI, 12.22% to 19.52%), but a decrease in precipitation also significantly increased MBC by 6.62% (95CI, 2.38% to 10.85%), and there was a significant difference between the two responses (*p* < 0.05) (Figure 2a).

Subgroup analyses revealed that slight and moderate precipitation increases significantly enhanced MBC, especially in grasslands, while extreme increases had no significant effects (Figure 2b). Moderate precipitation reduction in forests and slight reduction in grasslands suppressed MBC, but moderate reduction in grasslands unexpectedly stimulated MBC (Figure 2c). Short- and long-term precipitation addition increased MBC, particularly in grasslands, whereas short- and medium-term reductions decreased MBC in forests but increased it under short-term reduction in grasslands. Long-term reduction had no significant effect on MBC (Figure 2d,e).

For MBN, precipitation addition led to an overall increase of 15.56% (95% CI, 6.73% to 24.39%), while reduction caused a 10.89% (95 CI, −19.36% to −2.42%) decrease (*p* < 0.05) (Figure 2f). In grasslands, an increase in slight precipitation significantly raised MBN by 21.21% (95CI, 6.15% to 36.27%) (Figure 2g). In forests, the reduction in slight and moderate precipitation led to a decrease of 13.97% (95CI, −20.89% to −7.05%) and 24.06% (95CI, 30.08% to 18.04%), respectively, in MBN. In grasslands, a reduction in slight precipitation significantly decreased MBN by 16.45% (95CI, −29.87% to −3.02%), while a reduction in moderate precipitation significantly increased MBN by 14.44% (95CI, 6.54% to 22.34%) (Figure 2h). Long-term precipitation reduction significantly decreased MBN overall and in forests, while short- and medium-term changes had no significant effects in grasslands (Figure 2i,j).

### 3.2. Effects of Precipitation Changes on Soil Bacterial and Fungal Diversity in Forest and Grassland Ecosystems

Overall, bacterial diversity was not significantly affected by any form of precipitation change, including increases or decreases in amount, as well as variations in intensity (slight, moderate, or extreme) or duration (short-, medium-, or long-term) (Figure 3a,b). In contrast, fungal diversity decreased significantly under an extreme precipitation reduction, with an average decline of 6.67% (95% CI, −13.02% to −0.32%), while other treatments—including increases in all levels of precipitation and slight to moderate decreases in precipitation—had no significant effects (Figure 3c). Similarly, short-term precipitation reduction led to a 10.95% decrease in fungal diversity (95% CI, −21.38% to −0.51%), whereas medium- and long-term reductions, as well as all durations of precipitation increase, showed no significant impacts (Figure 3d).

### 3.3. Effects of Precipitation Changes on Soil C-, P-, and N-Cycling Enzyme Activities and Antioxidant Enzymes in Forest and Grassland Ecosystems

In grassland ecosystems, precipitation reduction significantly decreased the activity of soil carbon-cycling enzymes by 18.3% (95% CI, −34.00% to −0.26%), whereas no significant effects were observed under either precipitation increase or reduction at the overall level or within forest ecosystems (Figure 4a). For phosphorus-cycling enzymes, precipitation reduction led to a significant overall decline of 27.72% (95% CI, −40.86% to −14.58%), while precipitation increase had no significant effect. In grasslands specifically, precipitation increase elevated P-cycling enzyme activity by 15.24% (95% CI, 5.60% to 24.89%), and precipitation reduction decreased it by 14.68% (95% CI, −25.3% to −4.0%), with a significant difference between the two treatments (*p* < 0.05) (Figure 4b). Nitrogen-cycling enzyme activity was not significantly affected by either precipitation increase or reduction. However, antioxidant enzyme activity declined by 8.45% (95% CI, −16.13% to −0.77%) under precipitation reduction (Figure 4c).

### 3.4. Relationships Between Environmental Factors and Microbial Biomass, Diversity, and Enzyme Activity

The response of MBC to precipitation reduction was significantly negatively correlated with MAT (*p* < 0.05, Figure 5a), while its response to precipitation addition was positively correlated with MAP (*p* < 0.05, Figure 5b). In addition, MBC response to precipitation reduction showed a highly significant positive correlation with elevation (*p* < 0.001, Figure 5c). Bacterial diversity response to precipitation addition was negatively correlated with MAP (*p* < 0.05, Figure 5d), and its response to precipitation reduction was strongly positively correlated with elevation (*p* < 0.001, Figure 5e). Carbon-cycling enzyme activity response to precipitation addition was positively correlated with MAT (*p* < 0.05, Figure 5f), while nitrogen-cycling enzyme activity response to precipitation reduction also showed a significant positive correlation with MAT (*p* < 0.05, Figure 5g).

## 4. Discussion

### 4.1. Effects of Precipitation Changes on MBC and Nitrogen in Forest and Grassland Ecosystems

#### 4.1.1. Effects of Precipitation Changes, Including Their Magnitude and Duration, on MBC in Forest and Grassland Ecosystems

Field studies consistently show that increased precipitation enhances MBC across various ecosystems [48,49]. Our meta-analysis corroborates this general trend while revealing notable differences between forests and grasslands. In forest ecosystems, precipitation reduction significantly suppressed MBC, whereas precipitation addition produced no significant effects—a pattern consistent with findings of microbial biomass decline under drought but minimal response to increased water availability [50]. In contrast, grassland ecosystems showed a 15.87% increase in MBC under precipitation addition, which aligns with findings that attribute this stimulation to alleviated water limitations and enhanced substrate availability [51]. Interestingly, a 6.62% increase in MBC under precipitation reduction was also observed in grasslands, suggesting a degree of microbial compensation under drought stress. This resilience has been observed in semi-arid and Mediterranean grasslands, where microbial traits remained stable or even improved under reduced rainfall, likely due to inherent adaptation to fluctuating moisture conditions [52,53]. Further supporting this resilience, studies in desert steppe ecosystems have shown that reduced precipitation can unexpectedly enhance soil nutrient accumulation (e.g., SOC, TN, TP) and microbial biomass. This occurs due to reduced litter decomposition and limited nutrient leaching, which collectively enhance soil nutrient retention [54]. Our subgroup analysis further revealed that mild to moderate increases in precipitation significantly enhanced MBC in grasslands, with increases of 17.10% and 12.31%, respectively. In contrast, heavy precipitation increases did not significantly affect MBC in either ecosystem. This aligns with global syntheses and long-term experiments, suggesting that excessive water can reduce oxygen availability and subsequently constrain microbial metabolic activity [55,56].

In forests, moderate precipitation reduction consistently suppressed MBC, a result echoing findings from beech forests, where lower precipitation reduced both microbial biomass and soil carbon pools [57]. However, grasslands exhibited inconsistent responses to moderate reductions, as follows: some cases showed the stimulation of MBC, possibly due to compensatory microbial responses or shifts toward drought-adapted microbial taxa. These observations are supported by studies in semi-arid grasslands that report enhanced microbial carbon dynamics during recovery phases following moderate drought [58,59].

Importantly, our results, together with evidence from long-term drought experiments, suggest that extreme reductions in precipitation do not universally suppress MBC in grasslands. For instance, microbial biomass remained stable over five years of extreme drought in both desert and grassland environments, and only minor fluctuations were observed under simulated centennial-scale droughts [60,61]. Collectively, these findings align with broader patterns where environmental drivers modulate microbial responses to precipitation changes through both direct and indirect pathways. Increased water availability may indirectly enhance microbial diversity by promoting plant diversity and altering substrate inputs, whereas mild drought conditions often exert limited effects on microbial biomass in ecosystems where communities are adapted to periodic water stress or where favorable soil texture (e.g., sandy soils) buffers moisture fluctuations. For instance, in temperate oak forests, microbial biomass remained stable under experimental throughfall reduction, demonstrating drought tolerance until soil moisture dropped below a critical threshold (~10 vol%), which significantly suppressed microbial activity [62,63]. Microbial responses to increased precipitation, however, appear to be more transient. MBC and microbial abundance typically increase within the first five years of water addition before leveling off. In forest ecosystems, we observed that short- and medium-term precipitation additions had a minimal impact on MBC. This is consistent with canopy-level experiments showing that water augmentation alone does not significantly alter microbial biomass unless accompanied by other factors such as nitrogen deposition [64].

#### 4.1.2. Effects of Precipitation Changes, Along with Their Magnitude and Duration, on MBN in Forest and Grassland Ecosystems

In forest ecosystems, precipitation reduction significantly suppressed MBN, likely due to decreased soil moisture and extractable organic carbon (EOC), both of which limit microbial biomass and the abundance of functional genes [65]. Conversely, increased precipitation had little impact on MBN in forests. Even under doubled rainfall conditions, microbial biomass and nitrogen mineralization processes remained largely unchanged, possibly due to coarse soil textures and the presence of pre-adapted microbial communities [66]. In grassland ecosystems, MBN exhibited stronger positive responses to precipitation addition. Intensified rainfall has been shown to enhance microbial nitrogen uptake by increasing nitrate and ammonium assimilation, driven by improved water and nitrogen availability [67]. Under precipitation reduction, MBN in grasslands remained relatively stable, indicating functional resilience. Although drought can alter nitrogen-cycling gene abundance and reduce microbial biomass, labile nitrogen pools often remain unaffected, suggesting microbial systems adapted to water stress [68,69]. In forests, however, MBN declined significantly under even mild drought, with reductions of 13.97% and 24.06% under mild and moderate treatments, respectively. This finding aligns with studies in Mediterranean forests showing that seasonal drought leads to declines in microbial nitrogen pools [70]. On a global scale, MBN is highly sensitive to drought, with meta-analyses reporting an average reduction of 10.4%, suggesting that microbial activity responds more strongly to water availability than to nutrient levels [71]. Furthermore, mild drought has been shown to significantly suppress MBN in grasslands, with stronger effects seen as drought intensity and duration increase [72].

Unexpectedly, moderate precipitation reduction in grasslands led to a 14.44% increase in MBN. This may result from enhanced fine root turnover and rhizosphere activity, which improve nitrogen substrate availability and stimulate microbial nitrogen transformation [73]. In contrast, forest ecosystems consistently showed suppressed microbial nitrogen dynamics under both short- and long-term drought. For instance, short-term drought reduced biological nitrogen fixation by cyanobacteria in moss systems, despite stable cyanobacterial abundance [74]. Long-term drought altered the structure of diazotrophic bacterial communities in Mediterranean forests, possibly indicating functional shifts rather than compensation [75]. Grassland systems appear more resistant to drought duration. Our meta-analysis found no significant relationship between drought length and MBN in grasslands, likely due to characteristics such as shallow rooting, low productivity, and moisture-adapted microbial taxa.

### 4.2. Effects of Precipitation Changes on the Diversity of Bacterial and Fungal Communities in Soil Microbial Populations in Forest and Grassland Ecosystems

#### 4.2.1. Effects of Precipitation Changes, Including Their Magnitude and Duration, on Bacterial Communities in Forest and Grassland Ecosystems

Bacterial diversity showed high stability across different precipitation treatments, regardless of direction, intensity, or duration. This insensitivity aligns with findings from both forest and grassland ecosystems, where even 30% to 75% changes in precipitation failed to alter bacterial diversity metrics. In Korean pine forests, bacterial diversity remained unchanged despite 30% rainfall variation, likely due to the physiological flexibility and resilience of dominant taxa across soil depths [76]. Similarly, alpine grassland studies reported no significant changes in bacterial Shannon diversity under precipitation shifts up to ±75%, attributed to drought-tolerant taxa such as Actinobacteria and the buffering effects of microbial network structure [77]. Moreover, bacterial diversity tends to be less responsive than fungal diversity under precipitation variation in subtropical forests. This is likely due to bacteria’s larger species pool buffering non-extreme drought stress versus fungi’s significant diversity reductions [78].

In addition to ecological explanations, methodological biases in DNA-based analyses may have contributed to the observed lack of bacterial diversity responses to precipitation changes. DNA extraction methods vary in their efficiency to lyse different microbial taxa and recover DNA from complex soil matrices, leading to inconsistencies in diversity estimates across studies [79,80]. Such protocol-driven variation can obscure real treatment effects, especially if sensitive taxa (e.g., Actinobacteria) are underrepresented by certain methods. Changey et al. further showed that extraction protocols can alter the detectability of environmental effects, even in long-term experiments [81]. In our meta-analysis, the integration of studies with heterogeneous methods likely introduced noise that masked subtle shifts in bacterial diversity. Standardizing extraction protocols is essential to improve cross-study comparability in future research.

#### 4.2.2. Effects of Precipitation Changes, Including Their Magnitude and Duration, on Fungal Communities in Forest and Grassland Ecosystems

Fungal diversity responded more sensitively to precipitation reduction than bacterial diversity, but only under specific conditions. Our meta-analysis revealed a significant 6.67% decline in fungal diversity under extreme precipitation reduction, whereas moderate and mild reductions, as well as all levels of precipitation increase, showed no consistent effects. This pattern is consistent with findings in upland heathlands, where summer drought reduced fungal richness and altered community structure due to limited plant-derived labile carbon, which constrains key fungal taxa such as phenol oxidase producers [82]. In contrast, moderate and light droughts had limited effects, likely due to the persistence of dormant fungal propagules that rapidly reestablish once moisture conditions improve. This compositional buffering underlies the observed resilience of fungal diversity to moderate precipitation changes [83]. Additionally, fungal diversity in bulk soil often remains stable under moisture stress, likely because soil matrices provide spatial buffering and because many fungi are less dependent on plant hydraulic status than root-associated microbes. These protective mechanisms help maintain diversity even during drought [84].

Short-term drought, however, caused notable reductions in fungal diversity. Multi-site experiments link this decline to community reordering and the potential loss of rare taxa, driven by drought-induced changes in root traits and soil nutrient dynamics [85]. Long- and mid-term droughts, as well as all forms of increased precipitation, had little effect, highlighting a temporal threshold beyond which fungal diversity stabilizes. Environmental factors such as soil pH, C/N ratio, and organic carbon content may exert stronger control on fungal community composition than precipitation alone [86]. Additionally, the dominance of drought-adapted taxa like dark septate endophytes (DSE) may further buffer fungal communities against moisture fluctuations. This has been observed in drylands and tallgrass prairies, where fungal diversity remained unchanged under long-term climate manipulation, likely due to ecological plasticity of dominant fungal groups [87,88].

### 4.3. Effects of Precipitation Changes on the Activities of Soil Carbon-, Phosphorus-, and Nitrogen-Cycling Enzymes and Antioxidant Enzymes in Forest and Grassland Ecosystems

#### 4.3.1. Effects of Precipitation Changes on the Activity of Soil Carbon-Cycling Enzymes in Forest and Grassland Ecosystems

Our meta-analysis showed that precipitation reduction significantly suppressed the activity of carbon-cycling enzymes in grassland ecosystems, with an average decline of 18.3%. This decline is primarily attributed to increased microbial carbon limitation and reduced carbon use efficiency under drought, resulting from decreased plant photosynthesis and reduced belowground carbon allocation [89]. Under such conditions, microbes prioritize maintenance respiration over growth, and the combined limitations of water and nutrients further restrict enzyme production. In contrast, carbon-cycling enzyme activity in forest ecosystems showed no consistent response to either increased or decreased precipitation, suggesting a degree of functional stability. Field studies in post-harvest forests reported that soil extracellular enzymes are less responsive to precipitation changes, possibly due to stronger abiotic controls and inherent buffering capacity within microbial communities [90,91]. This differential response indicates that microbial carbon metabolism in grasslands is directly constrained by soil moisture and substrate availability, whereas forests maintain enzymatic stability through long-term acclimatization to seasonal moisture variability.

#### 4.3.2. Effects of Precipitation Changes on the Activity of Soil Phosphorus-Cycling Enzymes in Forest and Grassland Ecosystems

In grassland ecosystems, precipitation increase significantly promoted the activity of phosphorus-acquiring enzymes. This enhancement is primarily driven by intensified phosphorus limitation under elevated water and nitrogen availability, which stimulates phosphatase secretion by both microbes and plant roots [92]. Improved soil moisture also facilitates substrate diffusion and stabilizes enzyme molecules, thereby enhancing phosphorus turnover across soil layers [93].

Conversely, precipitation reduction significantly suppressed phosphorus-cycling enzyme activity. Drought-induced declines in soil moisture reduce substrate mobility and microbial energy status, directly constraining enzyme synthesis. In particular, acid phosphatase activity shows strong sensitivity to drought, regardless of vegetation type, indicating that phosphorus acquisition processes are highly water-dependent [94]. Reduced plant biomass and organic input under drought conditions may also lead to the microbial downregulation of enzyme production as a resource-saving strategy, contributing to the observed reductions in phosphorus-transforming enzyme activity [95].

These findings suggest that soil phosphorus cycling in grasslands is highly responsive to changes in water availability, with increased precipitation stimulating and reduced precipitation constraining microbial enzymatic function.

#### 4.3.3. Effects of Precipitation Changes on the Activities of Soil Nitrogen-Cycling Enzymes and Antioxidant Enzymes in Forest and Grassland Ecosystems

Nitrogen-cycling enzymes exhibited no significant response to precipitation changes in either forest or grassland ecosystems. This aligns with field evidence suggesting that microbial nitrogen acquisition processes are less responsive to moderate hydrological shifts, likely due to their regulation by substrate stoichiometry rather than soil moisture alone [96]. These enzymes may possess limited physiological plasticity and thus maintain stable activity under variable precipitation. This stability likely stems from their regulation by substrate stoichiometry rather than hydrological variability. While microbial biomass may increase with improved water availability, enzyme production remains constrained when nitrogen is not limiting, reflecting a conservative resource allocation strategy [97]. This decoupling is consistent with evidence that microbial nutrient acquisition is governed more by elemental imbalances—particularly C:N ratios—than by biomass size or moisture fluctuations [98]. Moreover, enzyme activities have been shown to resist change even under dry–rewetting cycles, as microbes prioritize osmotic regulation over enzymatic investment when faced with moisture stress [99]. Together, these findings suggest that stoichiometric homeostasis buffers nitrogen acquisition enzyme activity against moderate precipitation shifts, explaining their stability despite concurrent biomass responses.

In contrast, antioxidant enzyme activities showed consistent sensitivity to precipitation reduction. Drought significantly suppressed oxidative and hydrolytic enzymes such as urease, protease, and β-glucosidase, primarily due to reduced soil water content and secondarily due to declines in substrate quality [100]. Under more extreme drying, reductions in enzyme activity were even more pronounced, with reported reductions exceeding 50%, attributed to microbial mortality and reduced enzymatic synthesis under prolonged stress [101].

Beyond substrate limitation and microbial mortality, the reduction in antioxidant enzyme activity under precipitation decrease also reflects oxidative stress and microbial metabolic adjustment. The drought-induced accumulation of reactive oxygen species (ROS) damages enzyme structure and function [102], while microbes downregulate energy-intensive antioxidant enzyme synthesis in favor of non-enzymatic ROS scavengers and osmolytes [103]. This trade-off prioritizes cell survival over enzymatic defense, particularly under prolonged stress. Symbiotic microbes like AMF further modulate antioxidant responses and alleviate drought severity through enhanced water uptake and gene-level regulation [104].

These results indicate that while nitrogen-cycling processes are relatively resistant to precipitation changes, antioxidant defense mechanisms in microbial communities are more vulnerable to drought stress and can serve as sensitive indicators of soil biochemical responses to drying conditions.

### 4.4. Relationship Between Environmental Factors and the Biomass, Diversity and Activity of Soil Microbial Communities

#### 4.4.1. The Relationship Between Environmental Factors and Soil Microbial Biomass

Our results reveal that MAT, MAP, and elevation significantly modulate MBC responses to altered precipitation. Specifically, MBC responses to precipitation reduction were negatively correlated with MAT, indicating that warmer climates may exacerbate microbial vulnerability under drought. Elevated MAT has been shown to restrict microbial biomass via diminished carbon inputs from root exudates and litter while simultaneously exacerbating drought-imposed constraints on respiration and substrate supply [105].

In contrast, MBC responses to increased precipitation showed a positive correlation with MAP. This suggests that regions with higher baseline water availability support more robust microbial growth under additional moisture input. Previous studies by Wichern and Joergensen and Montiel-González et al. confirm that greater MAP enhances microbial biomass and nutrient pools through improved nutrient diffusion, higher dissolved organic matter input, and a shift toward growth-oriented microbial strategies [106,107].

Elevation was also strongly positively correlated with MBC responses to drought, potentially due to enriched substrate availability and microbial adaptation in high-altitude environments. High-elevation soils often contain more organic matter and promote greater carbon use efficiency, helping microbes maintain biomass even under reduced moisture [108]. Building on this, the enhanced microbial resilience at elevation likely stems from a combination of greater substrate supply, improved moisture retention, and metabolic adaptation. Cooler temperatures reduce organic matter decomposition, increasing labile carbon and nitrogen pools that support microbial activity during drought [109,110]. In parallel, higher precipitation, snowmelt, and lower evapotranspiration preserve soil water, especially in deeper layers. These conditions, alongside elevated microbial carbon use efficiency and fungal-dominated communities, promote resource-efficient growth and sustain microbial biomass under moisture limitation [111].

#### 4.4.2. The Relationship Between Environmental Factors and Soil Microbial Diversity

Environmental gradients, particularly MAP and elevation, modulate microbial diversity responses to precipitation shifts, but their effects exhibit strong context dependence. The observed negative correlation between bacterial diversity response and MAP under increased precipitation suggests that excessive moisture may reduce microbial niche heterogeneity and functional potential by homogenizing the soil environment. This pattern likely reflects suppressed expression of metabolic and stress-related genes in saturated soils, where oxygen limitation and resource uniformity constrain microbial differentiation [112]. However, high MAP does not universally suppress diversity. In some Mediterranean ecosystems, bacterial diversity remains stable across broad precipitation gradients, potentially due to vegetation buffering and substrate-driven filtering mechanisms that decouple diversity patterns from direct hydrological control [113].

In contrast, increasing elevation appears to amplify bacterial diversity responses to precipitation reduction. Higher-altitude environments often feature cooler climates, richer litter inputs, and more stable microhabitats, all of which can support diverse microbial assemblages even under drier conditions. Enhanced α-diversity in arid mountain systems has been linked to co-occurring increases in MAP, vegetation richness, and improved soil structure, which together buffer microbial communities against water stress [114]. Further supporting this, microbial diversity has been shown to follow a U-shaped distribution along elevational gradients, where mid- to high-altitude regions combine improved soil pH, greater vegetation cover, and moderated moisture levels—conditions that jointly promote microbial coexistence and resilience [115].

#### 4.4.3. The Relationship Between Environmental Factors and Soil Microbial Activity

Temperature emerged as a key modulator of microbial enzymatic activity under altered precipitation regimes. The positive correlation between MAT and enzyme responses to both carbon and nitrogen transformations suggests that warming enhances microbial metabolic potential by stimulating carbon use efficiency and substrate availability. In warmer soils, accelerated microbial turnover can increase the demand for extracellular enzymes involved in nutrient acquisition, particularly those targeting labile organic compounds. This may explain the stronger activation of enzymes like leucine aminopeptidase under combined heat and moisture increases [116].

However, the effects of MAT are likely ecosystem-specific, reflecting historical thermal adaptation of microbial communities. For instance, microbial enzymes in colder regions often exhibit higher temperature sensitivity, implying that while warming enhances activity overall, the magnitude of response depends on thermal history and baseline metabolic rates [117]. The influence of MAT on nitrogen-cycling enzymes further highlights context-dependent microbial strategies. In nutrient-rich warm environments, elevated temperature tends to favor copiotrophic microbes with fast growth and high enzyme investment, thereby increasing nitrogen turnover. Conversely, in colder or nitrogen-limited systems, microbial communities may allocate more resources to enzyme production to compensate for reduced decomposition rates, maintaining function under thermal constraint [118,119].

These findings indicate that microbial responses to precipitation cannot be decoupled from ambient climatic conditions. Warming not only modulates the biochemical potential of soil microbes but also reshapes their ecological strategies, influencing how enzymatic pathways respond to changing water availability.

It is worth mentioning that in addition to natural ecosystems, soil microbial communities are key regulators of agroecosystem resilience and sustainable productivity under climate variability. Microbial communities generally exhibit strong resilience and resistance to environmental stressors by maintaining their structural and functional stability, which in turn sustain nutrient cycling and organic matter decomposition under drought stress [120], and these may further reinforce system resistance to climatic extremes [121]. The functional roles of fungi and bacteria, driven by their rapid adaptive traits, are central to stabilizing crop yields and enhancing ecosystem multifunctionality across farmlands [122]. Consequently, targeted soil moisture management to encourage beneficial microbes can serve as an effective means of promoting agricultural resilience to climate change.

## 5. Conclusions

This meta-analysis provides a comprehensive assessment of how precipitation changes influence soil microbial biomass, diversity, and activity across forest and grassland ecosystems. Our findings demonstrate that increased precipitation significantly enhanced MBC and MBN on a regional scale, particularly in grassland ecosystems, while reduced precipitation exerted stronger negative effects in forests.

Precipitation-induced responses varied with treatment intensity and duration, with mild to moderate increases and long-term additions generally promoting MBC, and with moderate drought reducing microbial biomass, especially in forest systems.

Soil microbial diversity exhibited limited sensitivity to altered precipitation. Bacterial diversity remained largely unchanged across all precipitation treatments, whereas fungal diversity decreased significantly only under intense and short-term precipitation reductions.

Changes in soil microbial activity showed element-specific patterns. In grasslands, precipitation reduction significantly decreased the activities of carbon- and phosphorus-cycling enzymes, while phosphorus-cycling enzyme activity increased under precipitation addition. Nitrogen-cycling enzyme activities were largely unresponsive to precipitation changes, but antioxidant enzyme activity declined under reduced precipitation.

Environmental factors significantly modulated microbial responses to precipitation shifts. MAT, mean annual precipitation, and elevation influenced microbial biomass and diversity responses, with MAT negatively associated with MBC responses to drought and positively associated with enzyme activity responses. These results highlight the importance of both climatic context and biome-specific characteristics in shaping microbial functional responses to altered water regimes.

## Figures and Tables

**Figure 1 microorganisms-13-01915-f001:**
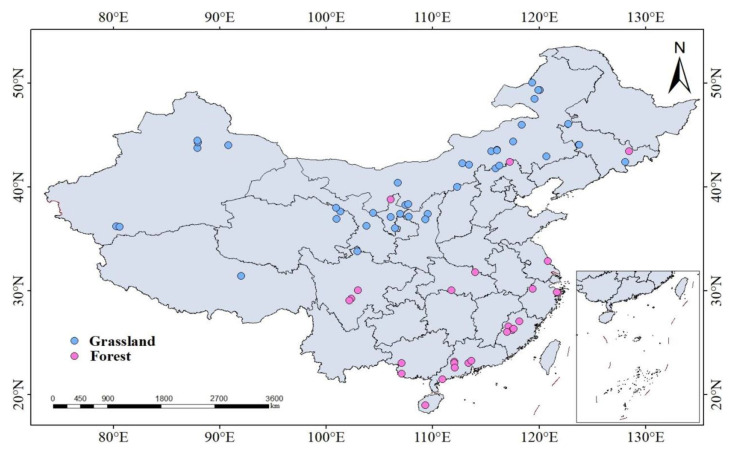
Location distribution of the study sites.

**Figure 2 microorganisms-13-01915-f002:**
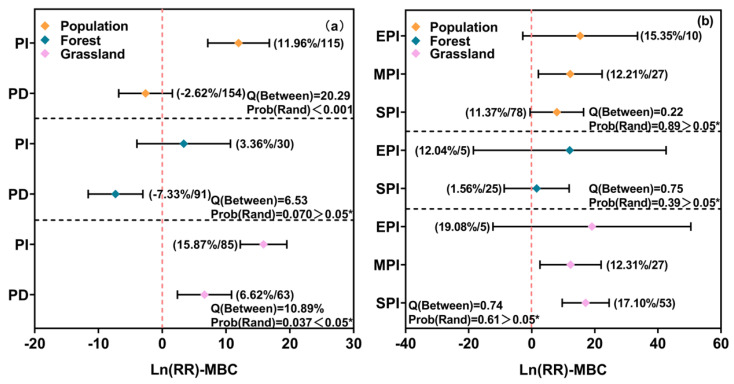
Independent and interactive effects of precipitation intensity and duration on MBC and MBN. PI: precipitation increase; PD: precipitation decrease; SPI: slight precipitation increase; MPI: moderate precipitation increase; EPI: extreme precipitation increase; SPD: slight precipitation decrease; MPD: moderate precipitation decrease; EPD: extreme precipitation decrease; STPI: short-term precipitation increase; MTPI: medium-term precipitation increase; LTPI: long-term precipitation increase; STPD: short-term precipitation decrease; MTPD: medium-term precipitation decrease; LTPD: long-term precipitation decrease. The error bars represent 95% CIs and indicate a significant effect when not overlapping with 0. Values in parentheses represent the effect size (Ln(RR)) and the number of observations included in each subgroup. Q(Between) and Prob(Rand) indicate the results of heterogeneity tests among subgroups, with lower *p*-values suggesting significant differences between groups. Asterisks indicate significance levels: * *p* < 0.05, ** *p* < 0.01.

**Figure 3 microorganisms-13-01915-f003:**
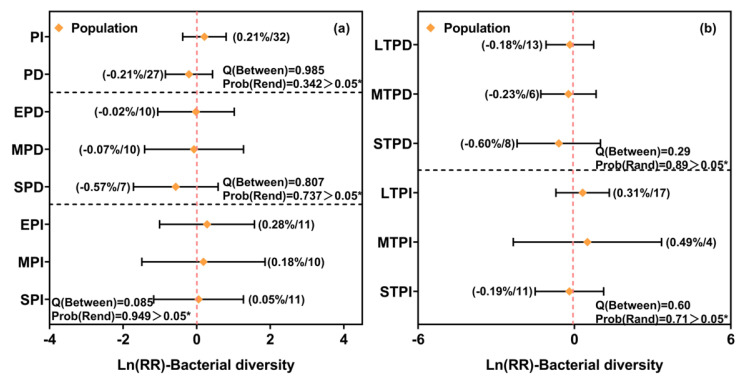
Independent and interactive effects of precipitation intensity and duration on bacterial and fungal diversity. PI: precipitation increase; PD: precipitation decrease; SPI: slight precipitation increase; MPI: moderate precipitation increase; EPI: extreme precipitation increase; SPD: slight precipitation decrease; MPD: moderate precipitation decrease; EPD: extreme precipitation decrease; STPI: short-term precipitation increase; MTPI: medium-term precipitation increase; LTPI: long-term precipitation increase; STPD: short-term precipitation decrease; MTPD: medium-term precipitation decrease; LTPD: long-term precipitation decrease. The error bars represent 95% CIs and indicate a significant effect when not overlapping with 0. Values in parentheses represent the effect size (Ln(RR)) and the number of observations included in each subgroup. Q(Between) and Prob(Rand) indicate the results of heterogeneity tests among subgroups, with lower *p*-values suggesting significant differences between groups. Asterisks indicate significance levels: * *p* < 0.05.

**Figure 4 microorganisms-13-01915-f004:**
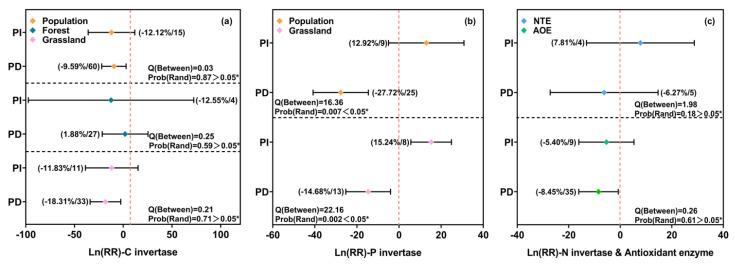
Effect sizes of soil C-, N-, and P-cycling and antioxidant enzyme activities in response to precipitation changes. PI: precipitation increase; PD: precipitation decrease; SPI: slight precipitation increase; MPI: moderate precipitation increase; EPI: extreme precipitation increase; SPD: slight precipitation decrease; MPD: moderate precipitation decrease; EPD: extreme precipitation decrease; STPI: short-term precipitation increase; MTPI: medium-term precipitation increase; LTPI: long-term precipitation increase; STPD: short-term precipitation decrease; MTPD: medium-term precipitation decrease; LTPD: long-term precipitation decrease. The error bars represent 95% CIs and indicate a significant effect when not overlapping with 0. Values in parentheses represent the effect size (Ln(RR)) and the number of observations included in each subgroup. Q(Between) and Prob(Rand) indicate the results of heterogeneity tests among subgroups, with lower *p*-values suggesting significant differences between groups. Asterisks indicate significance levels: * *p* < 0.05.

**Figure 5 microorganisms-13-01915-f005:**
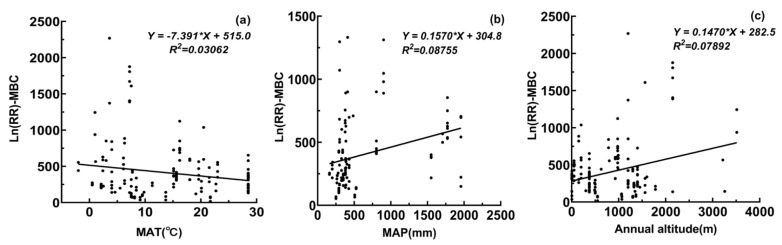
Responses of MBC, bacterial diversity, and C/N invertase activities to precipitation changes along gradients of MAT, MAP, and elevation.

## Data Availability

The data presented in this study are openly available in https://www.webofscience.com and https://www.cnki.net/.

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
