# Peer review of "Independent and Interactive Effects of Precipitation Intensity and Duration on Soil Microbial Communities in Forest and Grassland Ecosystems of China: A Meta-Analysis"

_microorganisms, 2025, doi:10.3390/microorganisms13081915_

Round 1
Reviewer 1 Report
Comments and Suggestions for Authors
The paper is a review on the effects of climatic factors on the proliferation of microbial species in various forest regions of China. To measure these variations, they integrated multiple factors, including precipitation levels, soil type, geographic altitude, and average environmental temperature. Their evaluated responses concluded, among other findings, an increase in bacterial and fungal biomass.
The introduction provides sufficient background information, clearly outlining the addressed problem. Thus, the references used (over 100) are appropriate for this type of work, reflecting the extensive data available for integration into the manuscript’s introduction.
No evidence of plagiarism or self-plagiarism was found in the references, confirming the originality of the presented work.
The study highlights several novel aspects, such as the variation in precipitation at different levels (light, moderate, and extreme by duration). It incorporates a wide diversity of data from multiple sources, including highly region-specific references for China, and considers various enzymatic activities to assess the impact of precipitation on the analyzed microbiota biomass.
The work holds scientific-academic relevance with notable contributions:
- It addresses spatial and temporal variability in microbial responses to hydrological changes.
- It revisits the asymmetry of responses between ecosystems (forests vs. grasslands) and microbial components (bacteria vs. fungi).
While similar themes have been reported elsewhere, this study maintains significant relevance. One major limitation, however, is its restriction to China despite the universal interest in the topic, making it regionally focused yet still innovative.
The figures and graphs are clear and align with the statistical evaluation methods, which reflect substantial effort in data compilation and classification, enhancing the work’s scientific value.
Suggestions for further discussion:
- Was expanding the literature search to include global regions beyond China not feasible, or was the focus intentionally limited?
- Were no data available for longer precipitation periods than those reported here?
- Were any analyses performed on microbial adaptation over time or extended periods?
- Bacterial and fungal classification is presented broadly, without taxonomic clarification. This could lead to divergent interpretations of the reported responses—specifying these classifications might strengthen the analysis.
Comments on the Quality of English Language
This writing is suitable for publication, requiring only a minor revision of style, spelling, and writing issues; other than that, I find it novel and of scientific interest.
Author Response
Comments1: Was expanding the literature search to include global regions beyond China not feasible, or was the focus intentionally limited?
Responses 1: We sincerely appreciate your insightful question regarding the geographic scope of our literature selection. The decision to focus exclusively on studies conducted within China was an intentional one, rather than a result of feasibility limitations. This regional focus was motivated by two key considerations. First, China presents a highly diverse set of forest and grassland ecosystems, with substantial geographic and climatic heterogeneity, making it an ideal context for analyzing biome-specific microbial responses to altered precipitation. Second, many relevant field experiments published in Chinese-language journals—particularly those involving nuanced precipitation manipulations—are not typically captured in global syntheses. By concentrating on China, we were able to integrate a large body of underrepresented data, thereby providing a more comprehensive and ecologically consistent analysis within a region of high relevance to climate change research. We fully agree that broader, cross-regional syntheses are valuable and necessary, and we hope our region-specific study can serve as a foundational step toward such future efforts.
Comments2: Were no data available for longer precipitation periods than those reported here?
Reviewer 2: Thank you for pointing this out. In our literature screening, we did identify a small number of studies with relatively extended experimental durations—up to 9 years in the longest case. To ensure analytical consistency and maintain adequate statistical power across subgroups, we followed precedent from earlier meta-analyses and classified precipitation duration into three categories: short-term (≤2 years), medium-term (2–4 years), and long-term (≥4 years).While we recognize the value of further stratifying long-term treatments, the availability of such extended field experiments is currently very limited. As a result, we were unable to conduct more detailed subgroup analyses without compromising statistical robustness.
Comments3: Were any analyses performed on microbial adaptation over time or extended periods?
Responses 3: We greatly appreciate your insightful suggestion regarding microbial adaptation over extended timeframes. While we recognize that microbial communities may exhibit adaptive or acclimatory responses to long-term environmental changes, the meta-analytic nature of our study posed certain limitations. Specifically, most of the primary studies included in our dataset did not report temporal dynamics or trajectory-based data that would allow for direct assessment of microbial adaptation over time. Moreover, due to variability in experimental durations and sampling frequencies across studies, it was not feasible to extract consistent time-series information for longitudinal analysis. However, we partially addressed this issue by categorizing precipitation treatments by duration (short-, medium-, and long-term), which allowed us to examine potential shifts in microbial responses across different temporal scales.
Comments 4: Bacterial and fungal classification is presented broadly, without taxonomic clarification. This could lead to divergent interpretations of the reported responses—specifying these classifications might strengthen the analysis.
Responses 4: We thank the reviewer for this valuable and constructive suggestion. We fully agree that incorporating finer-scale taxonomic information could enhance the ecological interpretation of microbial responses. However, due to the variability in microbial identification methods and sequencing resolution among the primary studies included in our meta-analysis, detailed taxonomic classifications (e.g., at the phylum or genus level) were not consistently reported across the dataset. In many cases, only aggregate diversity indices (e.g., Shannon index) were available, without accompanying compositional data. As such, we opted to focus our analysis on overall bacterial and fungal diversity responses to maintain consistency and ensure a sufficient sample size for robust statistical comparisons.
Reviewer 2 Report
Comments and Suggestions for Authors
General comments are in the pdf

Author Response
Comments 1: Re-adjust, the paragraph is very repetitive with connecting terms.
Response 1: Thank you for pointing this out. We have streamlined the repetitive content and excessive use of conjunctions. The revised passage is as follows: The following keywords were used for the literature search: China; Forest; Grassland; Precipitation increase/addition; Precipitation decrease/drought; Microbial biomass carbon (MBC); Microbial biomass nitrogen (MBN); Soil or microbial enzyme activity; Microbial diversity or Shannon index; Fungi or fungal community; Bacteria or bacterial community.
Comments 2: How many studies are recent?
Response 2: Thank you for pointing this out. There are 57 documents published in the past five years.The revised passage is as follows: Based on the above criteria, a total of 101 publications were collected, yielding 695 observations. Among these, 57 studies were published within the past five years.
Comments 3: What is the value of the point estimate of the meta-analysis result? How to know which information is heterogeneous?
Response 3: Thank you for pointing this out. "effect size, number of observations, Q(Between), Prob (Rand)" have been marked in Figures 2, 3, and 4.The revised passage is as follows: Values in parentheses represent the effect size (Ln(RR)) and the number of observations included in each subgroup. Q(Between) and Prob(Rand) indicate the results of heterogeneity tests among subgroups, with lower p-values suggesting significant differences between groups.
Comments 4: Please, standardize the abbreviations so that the information in the figures and text is clear.
Response 4: Thank you for pointing this out. All the terms "SMBC and SMBN" in the text have been unified as "MBC and MBN". And the abbreviation of "precipitation intensity and duration" was unified as: Precipitation intensity manipulations were categorized as slight (SPI/SPD, change within ±≤30%), moderate (MPI/MPD, ±30% to ±60%), and extreme (EPI/EPD, ±≥60%). Precipitation duration was classified into short-term (STPI/STPD, ≤2 years), medium-term (MTPI/MTPD, 2–4 years), and long-term (LTPI/LTPD, ≥4 years).
Comments 5: How? The statement is not clear What figure are you referring to?
Response 5: Thank you for pointing this out. The revised passage is as follows: In forests, the reduction of slight and moderate precipitation led to a decrease of 13.97%(95CI, -20.89% to -7.05%) and 24.06%(95CI, 30.08% to 18.04%) respectively in MBN. In grasslands, a reduction in slight precipitation significantly decreased MBN by 16.45%(95CI, -29.87% to -3.02%), while a reduction in moderate precipitation significantly increased MBN by 14.44%(95CI, 6.54% to 22.34%)(Fig. h).
Comments 6: Nothing was mentioned about the effects of the population that is shown in the Figure 2
Responses 6: Thank you for pointing this out. The population data is composed of both forest and grassland data. In the discussion section, we focus on forest and grassland ecosystems to examine how soil microbial communities in these two distinct ecosystems respond to changes in precipitation.
Comments 7: Form of cites should be corrected according to journal guide.
Responses 7: Thank you for pointing this out. Form of cites already be corrected according to journal guide.
Reviewer 3 Report
Comments and Suggestions for Authors
The manuscript Independent and Interactive Effects of Precipitation Intensity and Duration on Soil Microbial Communities in Forest and Grassland Ecosystems of China: A Meta-Analysis presents a timely and relevant investigation into the independent and interactive effects of precipitation intensity and duration on soil microbial communities across forest and grassland ecosystems in China. The study aims to address two fundamental and yet unresolved questions: 1. How do changes in precipitation intensity and duration independently and interactively influence soil microbial biomass, activity, and diversity across China’s forest and grassland ecosystems? 2. What roles do ecosystem-specific drivers play in shaping these responses under complex global change scenarios?
The paper is clearly structured and coherently written, with a logical organization of sections. The methodology is sound, and the results are effectively presented and discussed with reference to relevant and recent literature. The discussion is insightful, and the authors have succeeded in synthesizing the findings into a well-supported conclusion.
However, several minor revisions are recommended to further strengthen the manuscript: 1. Expansion of the Discussion: The section concerning the relationship between environmental factors and the biomass, diversity, and activity of soil microbial communities should be elaborated. Specifically, the interplay between temperature, moisture, soil properties, and nutrient availability with microbial responses needs further exploration. Additional references should be incorporated to reinforce the presented arguments and situate the findings within the broader scientific context. 2. Highlighting the Importance of Environmental Drivers: The discussion on the relationship between environmental factors and soil microbial activity should be expanded to underscore the ecological significance of these factors. Emphasis should be placed on their critical role not only in maintaining soil health and function but also in supporting sustainable agricultural productivity. This aspect is especially important given the increasing vulnerability of terrestrial ecosystems to climate variability. It is recommended that the authors incorporate additional references that discuss the implications of soil microbial dynamics for agroecosystem resilience and biogeochemical cycling.
In summary, this is a well-executed and highly relevant study that addresses pressing questions in the context of global environmental change. With minor revisions aimed at deepening the discussion and strengthening the theoretical grounding, the manuscript will make a valuable contribution to the field.
Author Response
Comments 1: Expansion of the Discussion: The section concerning the relationship between environmental factors and the biomass, diversity, and activity of soil microbial communities should be elaborated. Specifically, the interplay between temperature, moisture, soil properties, and nutrient availability with microbial responses needs further exploration. Additional references should be incorporated to reinforce the presented arguments and situate the findings within the broader scientific context.
Responses 1: The newly added discussion sections are as follows:
①Collectively, these findings align with broader patterns where environmental drivers modulate microbial responses to precipitation changes through both direct and indirect pathways. Increased water availability may indirectly enhance microbial diversity by promoting plant diversity and altering substrate inputs, whereas mild drought conditions often exert limited effects on microbial biomass in ecosystems where communities are adapted to periodic water stress or where favorable soil texture (e.g., sandy soils) buffers moisture fluctuations. For instance, in temperate oak forests, microbial biomass remained stable under experimental throughfall reduction, demonstrating drought tolerance until soil moisture dropped below a critical threshold (~10 vol%), which significantly suppressed microbial activity.
②Further supporting this resilience, studies in desert steppe ecosystems have shown that reduced precipitation can unexpectedly enhance soil nutrient accumulation (e.g., SOC, TN, TP) and microbial biomass. This occurs due to reduced litter decomposition and limited nutrient leaching, which collectively enhance soil nutrient retention.
Comments 2: Highlighting the Importance of Environmental Drivers: The discussion on the relationship between environmental factors and soil microbial activity should be expanded to underscore the ecological significance of these factors. Emphasis should be placed on their critical role not only in maintaining soil health and function but also in supporting sustainable agricultural productivity. This aspect is especially important given the increasing vulnerability of terrestrial ecosystems to climate variability. It is recommended that the authors incorporate additional references that discuss the implications of soil microbial dynamics for agroecosystem resilience and biogeochemical cycling.
Responses 2: The newly added discussion sections are as follows:
①Beyond natural ecosystems, soil microbial communities in grasslands are key regulators of agroecosystem resilience and sustainable productivity under climate variability. Environmental drivers such as precipitation regimes shape microbial structure and function, which in turn sustain nutrient cycling and organic matter decomposition during drought stress. Enhanced microbial biomass and enzyme activity under optimized water regimes—as observed in our grassland analyses—may further stabilize soil respiration and carbon sequestration, reinforcing system resistance to climatic extremes. These adaptive microbial traits, especially the functional roles of fungi and bacteria, directly support crop yield stability and ecosystem multifunctionality across agricultural landscapes. Thus, managing soil moisture to promote beneficial microbial communities offers a tangible strategy for climate-resilient agriculture.
Reviewer 4 Report
Comments and Suggestions for Authors
Brief Summary. This manuscript provides a comprehensive meta-analysis evaluating how variations in precipitation intensity and duration influence soil microbial biomass, diversity, and enzyme activity in forest and grassland ecosystems in China. Its key strength is the integration of a substantial dataset, clearly identifying differential responses of microbial parameters to increased and decreased precipitation, and highlighting ecosystem-specific patterns.
General Concept Comments
Strengths: The manuscript clearly identifies and addresses a significant gap in regional-level meta-analysis regarding soil microbial responses to altered precipitation patterns. A robust methodological framework and careful selection criteria are applied, ensuring a high-quality dataset. Clear differentiation between forest and grassland responses, adding valuable biome-specific insights.
Weaknesses: Some aspects of the discussion lack sufficient depth regarding potential mechanisms behind the observed differential responses between ecosystems. The interaction between microbial communities and plant dynamics (e.g., root exudation patterns under altered precipitation) is acknowledged but not explored in detail. The manuscript occasionally oversimplifies complex ecological interactions, potentially overlooking confounding factors like soil type variations and anthropogenic disturbances.
Specific Scientific Comments. Clearly stated hypotheses are presented (lines 104 - 107), and the meta-analysis provides a robust framework for testing these hypotheses across various precipitation treatments. The methodological approach (section 2.2 - 2.3) is appropriate and accurately applied. However, the classification thresholds for precipitation manipulation intensity (±≤30%, 30% < ± < 60%, ±≥60%) need more ecological justification or sensitivity analysis. Controls are clearly defined, but it would be to discuss whether the inclusion criteria for control conditions adequately capture baseline variability, especially given regional heterogeneity.
Specific Comments
Clearly written, but it would benefit from a mention of the number of studies or observations included to clarify the robustness of the findings immediately. (Lines 12 - 34)
Duration categories could be explained better - why is ≤2 years short-term and not, say, ≤1 year? (Line 135)
Figure 2 сlearly visualized; however, it would be beneficial to specify explicitly in the legend the exact number of observations per subgroup to give readers context on the strength of conclusions.
Figure 3. Clarify why bacterial diversity was consistently unresponsive - consider adding discussion on potential methodological biases (e.g., DNA extraction methods) that might have influenced these outcomes.
Figure 4. The marked reduction in antioxidant enzyme activity under precipitation decrease warrants additional mechanistic interpretation in the discussion - particularly regarding oxidative stress and microbial metabolic adjustments.
The relationships shown in Figure 5 need clearer ecological interpretation. It would be beneficial to expand on why elevation positively correlated with microbial biomass responses under drought - possibly relating to moisture and organic matter dynamics at higher elevations.
Y-axis scales and error bars should be standardized across figures to aid comparison.
Additional questions for the authors.
What was the rationale for choosing the specific thresholds for categorizing precipitation intensity (±≤30%, ±>60%) and duration (≤2, 2-4, ≥4 years)?
Why did nitrogen-cycling enzymes show limited responsiveness to precipitation changes, despite clear biomass responses in some cases?
Author Response
Comments 1: Clearly written, but it would benefit from a mention of the number of studies or observations included to clarify the robustness of the findings immediately.
Responses 1: Thank you for pointing this out. The revised passage is as follows: Here, using data exclusively from 101 field experiments conducted in China(yielding 695 observations), we investigated the impacts of altered precipitation on soil microbial biomass, diversity, and enzymatic activity in forest and grassland ecosystems.
Comments 2: Duration categories could be explained better - why is ≤2 years short-term and not, say, ≤1 year?
Responses 2: We appreciate your thoughtful suggestion. There is indeed no universally accepted standard for categorizing precipitation manipulation durations in ecological meta-analyses. To ensure both ecological relevance and balanced sample size across categories, we classified short-term precipitation treatments as those lasting ≤2 years. This threshold enabled a more even distribution of sample sizes across short-, medium-, and long-term groups among the 101 field studies included in our dataset.And precedents that roughly conform to this classification criterion have also been found. We have now clarified this rationale in the Materials and Methods section of the revised manuscript.The revised passage is as follows: Precipitation intensity manipulations were categorized as slight (SPI/SPD, change within ±≤30%), moderate (MPI/MPD, ±30% to ±60%), and extreme (EPI/EPD, ±≥60%). Precipitation duration was classified into short-term (STPI/STPD, ≤2 years), medium-term (MTPI/MTPD, 2–4 years), and long-term (LTPI/LTPD, ≥4 years). Since there is currently no universally accepted standard for categorizing precipitation manipulation intensity and duration, we defined thresholds based on the distribution of values in our dataset and their ecological relevance. The use of ≤2 years as the short-term cutoff helped balance sample sizes across categories. Although no unified scheme exists, similar classification logic has been adopted in previous ecological meta-analyses, supporting the general validity of our approach.
Comments 3: Figure 2 clearly visualized; however, it would be beneficial to specify explicitly in the legend the exact number of observations per subgroup to give readers context on the strength of conclusions.
Responses 3: Thank you for pointing this out. "effect size, number of observations, Q(Between), Prob (Rand)" have been marked in Figures 2, 3, and 4.The revised passage is as follows: Values in parentheses represent the effect size (Ln(RR)) and the number of observations included in each subgroup. Q(Between) and Prob(Rand) indicate the results of heterogeneity tests among subgroups, with lower p-values suggesting significant differences between groups.
Comments 4: Figure 3. Clarify why bacterial diversity was consistently unresponsive - consider adding discussion on potential methodological biases (e.g., DNA extraction methods) that might have influenced these outcomes.
Responses 4: Thank you for your insightful comment.We further explained at the methodological level the issue that bacteria produce insignificant responses in various dimensions of precipitation treatment.The newly added explanation paragraph is as follows: In addition to ecological explanations, methodological biases in DNA-based analyses may have contributed to the observed lack of bacterial diversity responses to precipitation changes. DNA extraction methods vary in their efficiency to lyse different microbial taxa and recover DNA from complex soil matrices, leading to inconsistencies in diversity estimates across studies. Such protocol-driven variation can obscure real treatment effects, especially if sensitive taxa (e.g., Actinobacteria) are underrepresented by certain methods. Changey et al. (2020) further showed that extraction protocols can alter the detectability of environmental effects, even in long-term experiments. In our meta-analysis, the integration of studies with heterogeneous methods likely introduced noise that masked subtle shifts in bacterial diversity. Standardizing extraction protocols is essential to improve cross-study comparability in future research.
Comments 5: Figure 4. The marked reduction in antioxidant enzyme activity under precipitation decrease warrants additional mechanistic interpretation in the discussion - particularly regarding oxidative stress and microbial metabolic adjustments.
Responses 5: Thank you for pointing this out. The newly added explanation paragraph is as follows: Beyond substrate limitation and microbial mortality, the reduction in antioxidant enzyme activity under precipitation decrease also reflects oxidative stress and microbial metabolic adjustment. Drought-induced accumulation of reactive oxygen species (ROS) damages enzyme structure and function, while microbes downregulate energy-intensive antioxidant enzyme synthesis in favor of non-enzymatic ROS scavengers and osmolytes. This trade-off prioritizes cell survival over enzymatic defense, particularly under prolonged stress. Symbiotic microbes like AMF further modulate antioxidant responses and alleviate drought severity through enhanced water uptake and gene-level regulation.
Comments 6: The relationships shown in Figure 5 need clearer ecological interpretation. It would be beneficial to expand on why elevation positively correlated with microbial biomass responses under drought - possibly relating to moisture and organic matter dynamics at higher elevations.
Responses 6: Thank you for pointing this out. The newly added explanation paragraph is as follows: Elevation was also strongly positively correlated with MBC responses to drought, potentially due to enriched substrate availability and microbial adaptation in high-altitude environments. High-elevation soils often contain more organic matter and promote greater carbon use efficiency, helping microbes maintain biomass even under reduced moisture.Building on this, the enhanced microbial resilience at elevation likely stems from a combination of greater substrate supply, improved moisture retention, and metabolic adaptation. Cooler temperatures reduce organic matter decomposition, increasing labile carbon and nitrogen pools that support microbial activity during drought. In parallel, higher precipitation, snowmelt, and lower evapotranspiration preserve soil water, especially in deeper layers. These conditions, alongside elevated microbial carbon use efficiency and fungal-dominated communities, promote resource-efficient growth and sustain microbial biomass under moisture limitation.
Comments 7: Y-axis scales and error bars should be standardized across figures to aid comparison.
Responses 7: Thank you very much for this thoughtful suggestion. We fully understand the importance of visual consistency in figures. However, in our forest plots, the Y-axis does not represent numerical values but categorical variables (e.g., treatment types such as PI, PD, SPI, etc.). These categories vary by subplot and are inherently distinct between figures. As such, applying a uniform Y-axis scale would not facilitate clearer comparison and may instead introduce visual clutter and misalignment with the ecological meaning of each grouping. We sincerely hope the reviewer will agree that, in this context, maintaining non-standardized Y-axis categories better preserves the clarity and interpretability of the figures.
Additional questions for the authors.
Question 1: What was the rationale for choosing the specific thresholds for categorizing precipitation intensity (±≤30%, ±>60%) and duration (≤2, 2-4, ≥4 years)?
Responses 1: We thank the reviewer for raising this important question. As there is currently no universally accepted standard for categorizing precipitation manipulation intensity and duration in ecological meta-analyses, we defined our thresholds based on both ecological relevance and the distribution of values across the compiled studies. Specifically, we used ±30% and ±60% as cutoffs to distinguish slight, moderate, and extreme precipitation changes—thresholds that capture increasing ecological stress and are consistent with classifications used in prior synthesis studies. For example, Yan et al. (2018) adopted similar intensity categories (<±30%, ±30–60%, >±60%) and divided treatment durations into ≤1 year, 1–5 years, and >5 years. While our thresholds differ slightly to improve sample balance and interpretability, they align conceptually with these published precedents, supporting the general validity of our classification framework.
Yan, G.; Mu, C.; Xing, Y.; Wang, Q. Responses and Mechanisms of Soil Greenhouse Gas Fluxes to Changes in Precipitation Intensity and Duration: A Meta-Analysis for a Global Perspective. Can. J. Soil Sci. 2018, 98 (4), 591–603. https://doi.org/10.1139/cjss-2018-0002.
Question 2: Why did nitrogen-cycling enzymes show limited responsiveness to precipitation changes, despite clear biomass responses in some cases?
Responses 2: Thank you for pointing this out. The newly added explanation paragraph is as follows: This stability likely stems from their regulation by substrate stoichiometry rather than hydrological variability. While microbial biomass may increase with improved water availability, enzyme production remains constrained when nitrogen is not limiting, reflecting a conservative resource allocation strategy. This decoupling is consistent with evidence that microbial nutrient acquisition is governed more by elemental imbalances—particularly C:N ratios—than by biomass size or moisture fluctuations. Moreover, enzyme activities have been shown to resist change even under dry-rewetting cycles, as microbes prioritize osmotic regulation over enzymatic investment when faced with moisture stress. Together, these findings suggest that stoichiometric homeostasis buffers nitrogen-acquisition enzyme activity against moderate precipitation shifts, explaining their stability despite concurrent biomass responses.
Reviewer 5 Report
Comments and Suggestions for Authors
This is an interesting meta-analysis of published research on the effects of environmental variables (especially precipitation) on significant eco-physiological variables of soil microorganisms in grasslands compared to forests at sampling sites in China. The reviewed studies are largely very recent or in the last decade. Overall, the manuscript is nicely organized and clearly written. A few recommendations for clarification or corrections are presented below.
My main suggestion to clarify their methodology of the meta-analysis is to provide more detail on how they calculated the natural log-transformed response ratio (ln RR) as a standardized effect size. The formula (1) in Lines 160 – 162 is
ln RR = ln Xt – ln Xc .
Specifically, can the authors please indicate what data was entered for Xt and what data was entered for Xc ? Also, a particular example would be very beneficial in making clear how they are using this as an index of treatment effect size. if applied as defined, it requires that data from two experimental conditions are available an experimental condition (Xt) and a control or other reference condition (Xc). Some additional explanation after lines 160-162 on exactly how this formula was applied would help the reader interpret the graphs in the Results figures where the ‘ln RR’ data is included on the abscissa, and the related graphical results are addressed in the text.
Specific recommendations
Throughout the manuscript, there are many places where a space is needed, either where sentences are not separated after a period, or between components within a sentence, for example separating parentheses from the surrounding text, etc.
For example Line 99, “---- interactions[40].Baseline climate----
Line 122, ---- following criteria:(1) The figures ----
Further suggestions.
Line Recommendation
39 It would help if the IPCC was defined the first time used – although it is a widely recognized entity. That is “ Global climate change is accelerating, and recent reports by the Intergovernmental Panel on Climate Change (IPCC) indicate that the rates------”
176 “Where Y represents the average effect size; yi denotes the effect size of an------"
The Y should have a bar above it and the yi should have the subscript i as shown here.
695 Reference (45) is incomplete. The proper complete entry is as follows:
Hedges, L. V.; Gurevitch, J.; Curtis, P. S. The meta‐analysis of response ratios in experimental ecology. Ecology 1999. 80 (4), 1150–1156.
Author Response
Comments 1: My main suggestion to clarify their methodology of the meta-analysis is to provide more detail on how they calculated the natural log-transformed response ratio (ln RR) as a standardized effect size. The formula (1) in Lines 160 – 162 is “ln RR = ln Xt – ln Xc” . Specifically, can the authors please indicate what data was entered for Xt and what data was entered for Xc ? Also, a particular example would be very beneficial in making clear how they are using this as an index of treatment effect size. if applied as defined, it requires that data from two experimental conditions are available an experimental condition (Xt) and a control or other reference condition (Xc). Some additional explanation after lines 160-162 on exactly how this formula was applied would help the reader interpret the graphs in the Results figures where the ‘ln RR’ data is included on the abscissa, and the related graphical results are addressed in the text.
Responses 1: Thank you for your helpful suggestion. In our revised manuscript, we clarified the meaning of Xt and Xc in the Materials and Methods section. Specifically, Xt represents the mean value of a microbial parameter (e.g., MBC, MBN, Shannon index, or enzyme activity) under a precipitation manipulation treatment (either increased or reduced precipitation), and Xc is the corresponding mean under ambient (control) conditions. These values were directly extracted from the original studies.
As an example, if a study reported MBC as 250 mg C kg-1 soil under increased precipitation (Xt) and 200 mg C kg-1 soil under ambient conditions (Xc), the response ratio would be calculated as lnRR = ln(250) - ln(200) ≈ 0.22. This corresponds to a 24.6% increase in MBC relative to the control. This transformation allows us to use a standardized effect size across studies with different scales and units, and is applied consistently across all indicators analyzed in this meta-analysis.
Comments 2: Throughout the manuscript, there are many places where a space is needed, either where sentences are not separated after a period, or between components within a sentence, for example separating parentheses from the surrounding text, etc.
Responses 2: Thank you for pointing this out. We have carefully reviewed the entire manuscript line-by-line to correct all instances of missing spaces, including those after periods/commas/colons, around parentheses/brackets, between numbers and units, and around mathematical operators. We believe these revisions significantly improve readability.
Comments 3: It would help if the IPCC was defined the first time used – although it is a widely recognized entity. That is “ Global climate change is accelerating, and recent reports by the Intergovernmental Panel on Climate Change (IPCC) indicate that the rates------”
Responses 3: Thank you for pointing this out. The revised passage is as follows: Global climate change is accelerating, and recent reports by the Intergovernmental Panel on Climate Change (IPCC) indicate that the rates of warming and sea-level rise in the 21st century are unprecedented over the past 2000 years.
Comments 4: “Where Y represents the average effect size; yi denotes the effect size of an------"
The Y should have a bar above it and the yi should have the subscript i as shown here.
Responses 4: Thank you for pointing this out. The revised passage is as follows: Where y represents the average effect size; yi denotes the effect size of an individual study.
Comments 5: Reference (45) is incomplete. The proper complete entry is as follows:Hedges, L. V.; Gurevitch, J.; Curtis, P. S. The meta‐analysis of response ratios in experimental ecology. Ecology 1999. 80 (4), 1150–1156.
Responses 5: Thank you for pointing this out. We have corrected the citation format of this reference.